rsob.royalsocietypublishing.org

Subject Area:
molecular biology/neuroscience

Keywords:
odorant-binding protein, olfaction, antenna, *Drosophila*, insect

Author for correspondence:
John R. Carlson
e-mail: john.carlson@yale.edu

†These authors contributed equally to this study.

# The diverse small proteins called odorant-binding proteins

Jennifer S. Sun†, Shuke Xiao† and John R. Carlson

Department of Molecular, Cellular and Developmental Biology, Yale University, New Haven, CT 06520, USA

 JSS, 0000-0002-4274-0504; JRC, 0000-0002-0244-5180

The term 'odorant-binding proteins (Obps)' is used to refer to a large family of insect proteins that are exceptional in their number, abundance and diversity. The name derives from the expression of many family members in the olfactory system of insects and their ability to bind odorants *in vitro*. However, an increasing body of evidence reveals a much broader role for this family of proteins. Recent results also provoke interesting questions about their mechanisms of action, both within and outside the olfactory system. Here we describe the identification of the first Obps and some cardinal properties of these proteins. We then consider their function, discussing both the prevailing orthodoxy and the increasing grounds for heterodox views. We then examine these proteins from a broader perspective and consider some intriguing questions in need of answers.

## 1. The original Obp

The first Obp was discovered in 1981 [1]. A sex pheromone from the silk moth *Antheraea polyphemus* was labelled and incubated with an extract from antennae of this species. The pheromone bound a small protein, approximately 15 kD, which was detected in the antenna of male moths, but not female moths. The protein was further localized to olfactory sensilla of the male antenna but was not found in other tissues. The protein was extremely abundant: approximately 15 μg in a single antenna. A variety of other moths were then examined, and each was found to contain a male-specific antennal protein of comparable size and abundance that bound the pheromone of *A. polyphemus* [2,3].

This protein, sometimes referred to in the literature as a pheromone-binding protein (Pbp), will be referred to here by the more general term of odorant-binding protein (Obp). We note that mammals also contain proteins called Obps, which were first identified in the nasal mucosa of cows [4–6]. These proteins, however, are structurally distinct from insect Obps and their functions are largely unknown.

## 2. Remarkable properties of Obps

Obps have been identified and characterized from a variety of insects [7,8], including *Drosophila* (figure 1) [9,10]. Obps are striking in several respects.

### 2.1. Number

When the *Drosophila* genome was sequenced, Obps were found to constitute a very large family. There are 52 Obps in *Drosophila* [11], comparable to the number of odour receptors (Ors) or gustatory receptors (Grs). Some other insects contain even more *Obp* genes: the malaria mosquito *Anopheles gambiae* contains 69 [7], and the German cockroach *Blattella germanica* contains 109 [12]. The *B. germanica* repertoire is the largest identified to date; its members were identified in TBLASTN searches and include not only 'Classic' OBPs with six cysteines, but also a number of variants that contain either four cysteines, called 'Minus-C', or

rsob.royalsocietypublishing.org   *Open Biol.* **8**: 180208

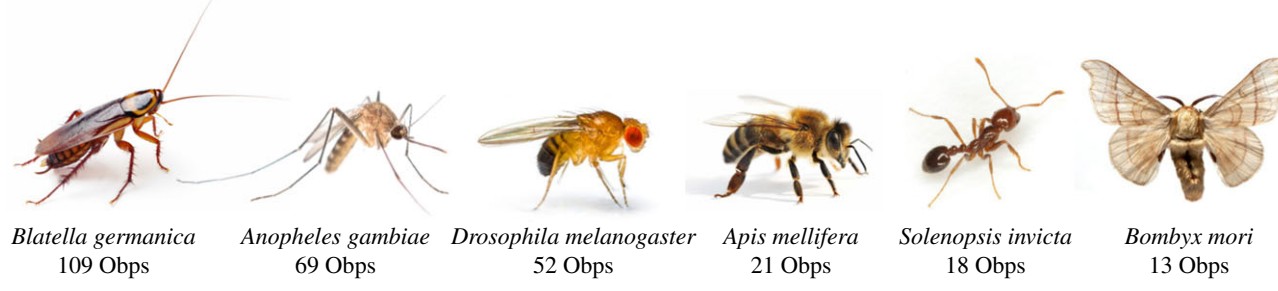

*Blatella germanica*
109 Obps

*Anopheles gambiae*
69 Obps

*Drosophila melanogaster*
52 Obps

*Apis mellifera*
21 Obps

*Solenopsis invicta*
18 Obps

*Bombyx mori*
13 Obps

**Figure 1.** Obps are numerous and widely distributed across insects. The number of annotated Obp genes is shown for a variety of insects.

eight cysteines, called 'Plus-C'. The *Drosophila* repertoire also includes such variant Obps, although none were among the highly expressed antennal Obps identified in a recent RNAseq analysis [13].

## 2.2. Abundance

Some Obps, such as the original Obp isolated in silk moths, are expressed at extremely high levels. Of the 10 most abundantly expressed genes in the olfactory segment of the *Drosophila* antenna, five are *Obps*, as judged by RNAseq analysis [13]. The most abundant of these RNAs are expressed at levels almost three orders of magnitude higher than that of a typical *Or* RNA.

## 2.3. Diversity

Obps are highly divergent in sequence. For example, among the *Drosophila* Obp family, members share only 20% amino acid identity on average [11,14]. Only two Obps have clear orthologues across a variety of insect orders examined [7,15]. However, Obps are similar in that they are typically small (approx. 14 kD) and generally contain six conserved cysteines [14].

## 3. Structure, expression and binding

The structure of an Obp from the silk moth *Bombyx mori* was determined in 2000 in two studies, one using X-ray crystallography and one NMR [16,17]. Structures of more than 20 Obps have subsequently been resolved. Obps typically contain six α-helices, three disulfide bridges and an internal cavity that may bind small hydrophobic molecules (figure 2). Crystallized Obps have revealed a dimeric structure with a binding pocket that consists of a tunnel extending into each of the two subunits [18,19].

The expression pattern of the first Obp within the moth antenna was examined at high resolution by immuno-electron microscopy [20]. Insect antennae contain several morphologically distinct classes of sensilla, including trichoid, basiconic and coeloconic sensilla. These sensilla are perforated by pores or channels through which odorant molecules can pass. Within the shaft of the hair are the dendrites of olfactory receptor neurons (ORNs), which are bathed in an aqueous sensillum lymph. At the base of the sensillum lie the cell bodies of the ORN and of three kinds of auxiliary cells: trichogen (shaft), tormogen (socket) and thecogen (sheath) cells (figure 3a).

Immuno-electron microscopy showed that the *A. polyphemus* Obp is localized to trichoid sensilla, which respond electrophysiologically to pheromone. The Obp is in the

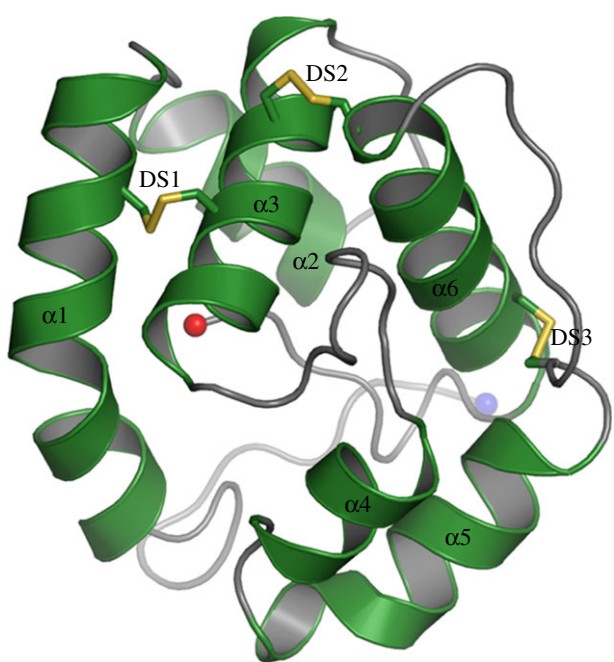

**Figure 2.** Structure of an Obp. Obp1 of *Aedes aegypti* is visualized with six α-helices (α1−6) and three disulfide linkages (DS1−3) labelled. Adapted from [18].

sensillum lymph (figure 3b,c). Obp was also observed in secretory organelles of the trichogen and tormogen cells, supporting a role for these cells in producing and secreting Obp into the lymph.

Subsequent expression analysis of a variety of insect Obps has revealed that different Obps are expressed in different morphological classes of sensilla. Moreover, some Obps are expressed in subsets of the sensilla of a particular morphological class [21]. This selective expression is of particular interest because different subsets of sensilla are functionally distinct, a pattern of organization that has been defined at high resolution in *Drosophila*.

*Drosophila* contains 10 different functional types of basiconic sensilla, named ab1 (antennal basiconic 1) to ab10, which respond electrophysiologically to different odorants. Each sensillum type contains up to four ORNs, typically two, that express distinct Ors in a stereotyped pattern. An extensive double-label analysis with *Obp* and *Or* markers produced an Obp-to-sensillum map [21]. The map showed that different basiconic sensillum types express different subsets of Obps, and individual Obps are expressed in various subsets of sensilla (figure 3d).

The original moth Obp was identified by virtue of its binding to a labelled pheromone. Many other Obps have

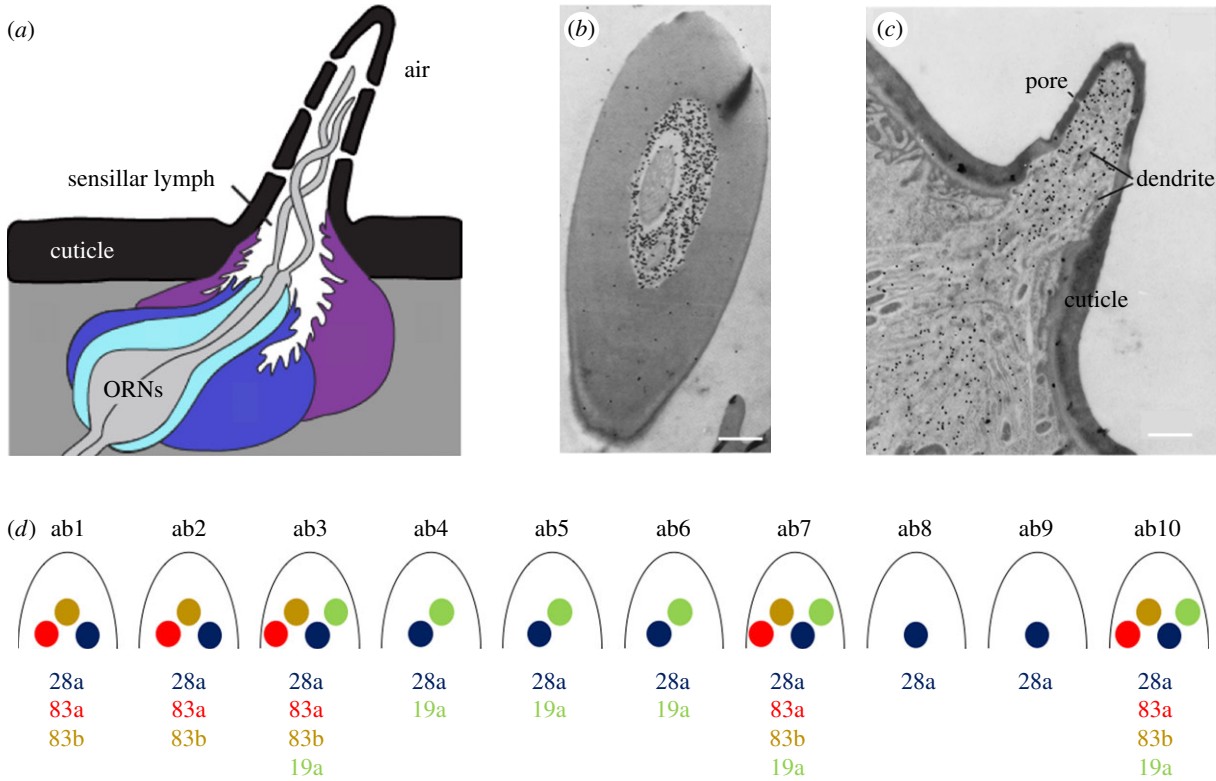

**Figure 3.** Expression of Obps in sensilla. (*a*) A typical olfactory sensillum. Auxiliary cells at the base of the sensillum are shown in blue and purple. Olfactory receptor neurons (ORNs) are in grey. Obps are synthesized and secreted into the sensillar lymph by auxiliary cells. The sensillum contains pores that allow odorants to enter and bind to olfactory receptors on the ORN dendrites. Adapted from [13,21,22]. (*b*) Immuno-electron microscopy of a cross-section from an *Antheraea polyphemus* trichoid sensillum, demonstrating the density of Obp labelling within the sensillar lymph. Scale bar, 1 μm. Adapted from [20]. (*c*) Immuno-electron microscopy of a longitudinal section from *Drosophila* trichoid sensilla labelled with anti-Obp83a. Scale bar, 1 μm. Adapted from [23]. (*d*) Map of highly abundant Obps expressed in antennal basiconic sensilla in *Drosophila*.

now been examined for their ability to bind a wide variety of compounds [8]. These *in vitro* binding studies have reported different degrees of affinity and selectivity for different Obps. Most Obps bind hydrophobic compounds [24]; some appear to bind a broad spectrum of hydrophobic ligands [25].

## 4. The prevailing orthodoxy

The findings that Obps are found in the sensillum lymph, and that they bind odorants *in vitro*, have led to a widely held and often cited model: that Obps bind, solubilize and transport odorants across the sensillum lymph to Ors in the dendritic membranes (figure 4). The model is attractive in that it offers a mechanism by which odorants, most of which are hydrophobic, can traverse an aqueous sensillum lymph. The model has been elaborated to include a step in which Obps change conformation and release their bound odorants upon interaction with negatively charged membranes of ORNs [26–28].

There is support for this model. The model predicts that the reduction of Obp function will reduce odorant transport and thereby reduce olfactory function. Consistent with this prediction, mutation of the *Obp76a* gene in *Drosophila* reduces the electrophysiological response of the sensilla that normally express it to the pheromone cis-vaccenyl acetate (cVA) [29]. RNAi knockdown of Obps in two mosquito species reduced the electroantennogram response—a measure of the summed response to a population of sensilla—to certain odorants [30,31]. RNAi knockdown of Obps in *Drosophila* reduced

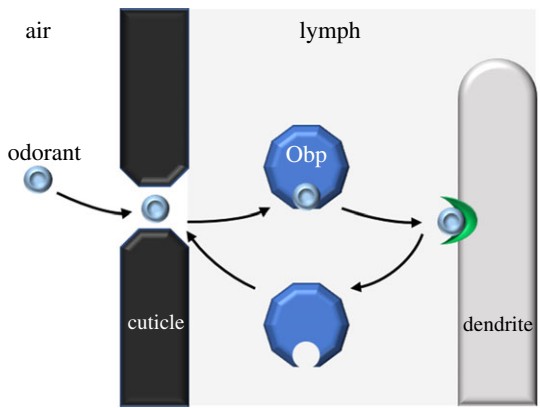

**Figure 4.** Standard model of Obp function in olfactory sensilla. The prevailing model of Obp function is that Obps bind odorants and carry them through the sensillar lymph to odour receptors in the membranes of ORN dendrites.

behavioural responses, in a sex-specific manner [32]. These results all indicate an abnormality in olfactory function. They do not, however, demonstrate that the loss of function is due to a failure of odorant transport.

## 5. Grounds for heterodoxy

Historically, other models have been proposed for the function of antennal Obps [6,8,24]. For example, the initial report describing the discovery of the first Obp proposed that it acted not in the initiation of odorant response but in

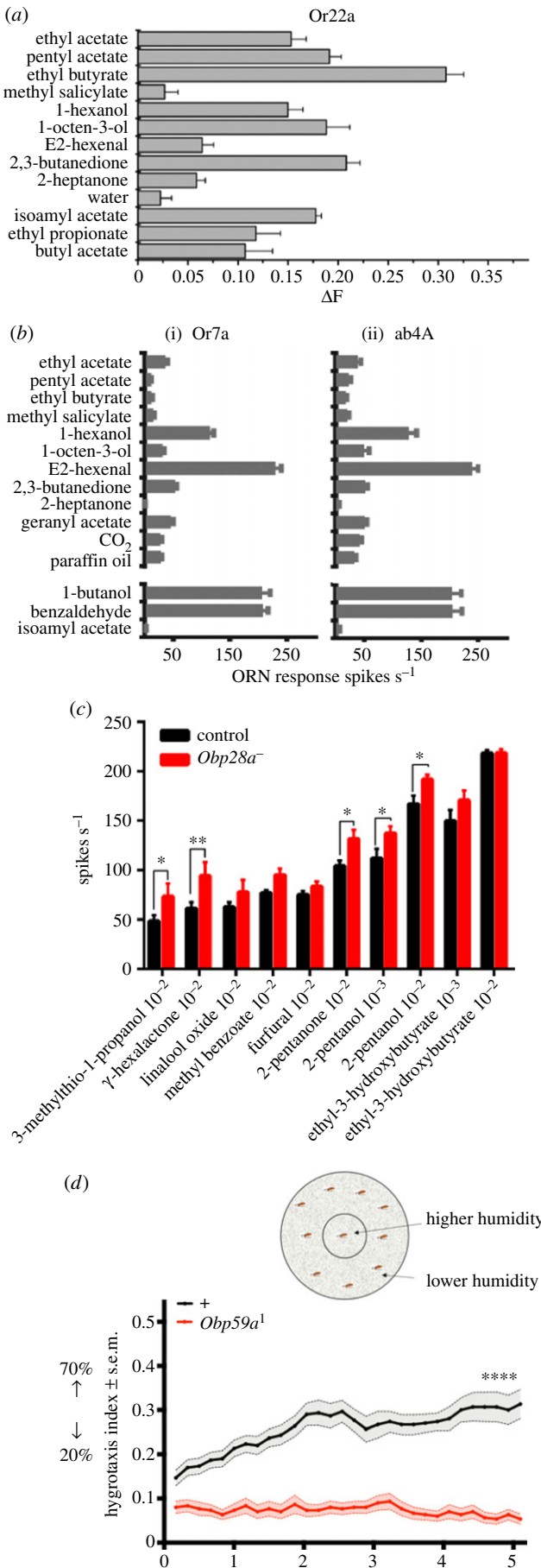

rsob.royalsocietypublishing.org  *Open Biol.* **8**: 180208

**4**

**Figure 5.** (*Opposite.*) Results not predicted by the standard model. (*a*) Or22a expressed in Sf9 cells responds to a variety of odorants in the absence of an Obp. Adapted from [33]. (*b*) Similarity of the response profile of Or7a in the *Drosophila* 'empty neuron' (i) to the profile of Or7a in the ab4A neuron (ii) in which Or7a is endogenously expressed. The profiles are very similar despite the different roster of Obps to which the two neurons have access. Adapted from [34]. (*c*) A mutant ab8 sensillum depleted of abundant Obps did not show a decrease in odorant response compared to a control ab8 sensillum; in fact, the mutant response is greater for several odorants. $*p < 0.05$, $**p < 0.01$. Adapted from [21]. (*d*) Mutant flies lacking Obp59a exhibit a reduced preference for regions of higher humidity. $****p < 0.0001$. Adapted from [35].

critical for insects that navigate towards a source of odorant and need a sensitive means of measuring instantaneous odorant concentration. Other roles that have been proposed for Obps include protection of odorants from degradative enzymes and filtering of odorants.

In considering the odorant transport model, which is now widely accepted, several experimental results should be taken into account. First, odour receptors expressed in heterologous systems such as Sf9 cells respond to odorants in the absence of Obps (figure 5*a*) [33,36]. Moreover, the response profiles of Ors *in vitro* are very similar to those *in vivo*, in at least some if not all cases. Second, when individual *Or* genes were misexpressed in a sensillum that contains a different complement of Obps than the sensillum in which the Or is endogenously expressed, the Ors conferred responses very similar to those in the endogenous sensillum, in most if not all cases (figure 5*b*) [34,37]. Thus, a unique complement of Obps is not essential in all cases for odorants to elicit a normal response *in vivo*. Third, there is some evidence that Obps increase the sensitivity of responses in an *in vitro* heterologous cell expression system [38], but in at least one experimental system the sensitivity could be increased equally by the addition of albumin [39].

Taken together, these results would seem to leave open the question of why olfactory organs express large, diverse families of Obps, in different patterns. The results suggest the possibility that the complexity of the Obp repertoire may serve a function other than odorant transport.

A powerful way of addressing the function of Obps is to examine the effect of eliminating them *in vivo*; however, a complication in this approach is that in most species the molecular organization of Obps has not been defined. It is unknown which Obps are expressed in which sensilla, nor is it known whether the elimination of one Obp would leave other Obps, possibly of redundant function.

The Obp-to-sensillum map in the basiconic sensilla of *Drosophila* (figure 3*d*) was constructed in part to address this issue [21]. It revealed that the ab8 sensillum contained a single highly expressed *Obp*, *Obp28a*. The *Obp28a* gene was then deleted, and the effects were analysed electrophysiologically.

Remarkably, elimination of the single highly expressed Obp from the ab8 sensillum did not reduce the magnitude of its responses to any of a wide variety of odorants (figure 5*c*) [21]. The simplest interpretation of these results is that the ab8 sensillum does not require an abundant Obp for odorant transport. Nor is Obp28a required for odorants to reach receptors in the ORN membranes. We note that experiments with long and intense odour stimuli did not reveal a role for Obp28a in odour clearance. Nor do the data support roles in filtering odorants or protecting them from degradative enzymes.

its termination [1]. An odorant would first activate a receptor, and then bind to the Obp, which would thereby act in clearing odorant from the sensillum lymph. Odorant clearing is

rsob.royalsocietypublishing.org  Open Biol. **8**: 180208

Interestingly, the magnitudes of the electrophysiological responses in *Obp28a* mutants were actually greater than in the control, for a number of odorants across a wide concentration range [21]. The greatest increase occurred during the initial phase of the response. It is possible that after a sudden influx of odorant into the sensillum, Obp28a binds some of the odorants, thereby reducing the amount available to activate receptors. After the termination of an odorant pulse, Obp28a might release odorant. In this manner, Obp28a might serve as a buffer against sudden changes in odour levels. However, it is also possible that Obp28a affects the response less directly, perhaps by affecting the sensillum or its physiology in other ways.

# 6. A broader perspective on Obps

Expression of the original Obp, and many others, is localized to the lymph of olfactory sensilla on the antenna, in the same compartment as the dendrites of ORNs. However, an increasing number of Obps have been found elsewhere.

First, Obp19d localizes to coeloconic sensilla of *Drosophila* that contain an inner and an outer compartment, of which only the inner compartment contains ORN dendrites [40]. Obp19d localizes to the outer compartment, however, and thus a role in odorant delivery to receptors seems unlikely.

Obp59a has recently been mapped to humidity-detecting sensilla of the *Drosophila* antenna [21,35]. Genetic analysis shows that it is required for hygroreception, an entirely different function than olfaction (figure 5*d*) [35]. Mutants lacking Obp59a are abnormal in their responses to humidity. Loss of Obp59a also increases desiccation resistance of flies. The molecular mechanism of hygroreception is not understood, but one prominent model is that a change in humidity alters the structure of hygroreceptive sensilla and that this alteration is transduced into a neural response [41]. It seems unlikely that Obp59a carries water molecules to a receptor; rather, another role, perhaps one in governing the structure or composition of the sensillum, seems more likely.

Obps are also expressed in the taste organs of various insects [42–46]. Obp49a is expressed in the labellum, the primary taste organ of the *Drosophila* head, where it has been found to act in the inhibition of sweet taste neurons by bitter compounds [45]. Obp57d and Obp57e are expressed in the leg, also a taste organ in fruit flies, and influence their host-plant preferences [46,47].

Other Obps are expressed far afield [48]. Obp6 of the tsetse fly is expressed in the larval gut, where it is upregulated by an obligate symbiotic bacterium [49]. This upregulation induces systemic expression of a transcription factor and the development of the fly's immune system. Other Obps have been localized to glands in a wide variety of insects, including sex pheromone glands of Lepidoptera and venom glands of wasps [50,51]. Still other Obps have been localized to male reproductive organs [52–54]. In most of these glands and organs, the molecular and physiological functions of the Obps remain speculative. However, these examples support the notion that Obps bind a variety of ligands other than odorants, and serve a variety of functions other than transport to odour receptors. In fact, recent RNAseq data in *Drosophila* have revealed the expression of more than 25 *Obps* in tissues outside the olfactory organs of the fly [55].

# 7. The great unknown

There is a massive amount of information about the expression of Obps, from a Noah's Ark of different insects, and about their binding to a wide diversity of compounds *in vitro*. But there is a great need for analysis of their functions *in vivo*, at the molecular, physiological and organismal levels.

At the molecular level, antennal Obps are clearly capable of binding small ligands, including many odorants and pheromones. However, binding affinities *in vitro* may not always reflect binding affinities *in vivo*, where conditions may differ. *In vitro* studies are extremely useful in making predictions about Obp function *in vivo*, but these predictions need to be tested. There are few cases in which knockdown or knockout of an Obp has been shown to reduce the electrophysiological response of a defined sensillum specifically to the odorants that bind the Obp *in vitro*.

Accordingly, a high priority for future research is to take advantage of the Obp-to-sensillum map generated in *Drosophila* so as to create a variety of 'empty sensilla' that lack abundant Obps. The effects of removing all abundant Obps can be studied in a number of different sensilla, and the effects of removing single Obps can be examined as well.

Another attractive research direction is to try to identify ligands *in vivo*. Some antennal Obps are expressed in regions devoid of ORN dendrites, suggesting that these Obps and perhaps others may bind ligands other than odorants [21,40]. Unbiased identification of Obp ligands, for example, by identifying ligands bound to Obps isolated directly from antennae, could reveal unexpected *in vivo* ligands.

Identification of *in vivo* ligands could inform two critical questions: What do Obps do after binding ligands, and what happens to the ligands? These questions may also be investigated by physiological analysis of defined sensilla whose Obp content has been manipulated in a defined way. Anatomical analysis of manipulated sensilla could also be informative, especially given the extremely high abundance of Obps. Could removal of Obps affect the ultrastructure of olfactory sensilla, such as the structure of their pores?

Finally, organismal roles of Obps are becoming discovered at an accelerating pace. In addition to hygroreception, desiccation resistance, taste and immune system development, a recent study has revealed a role in promoting male aggression for an antennal Obp of *Drosophila*, Obp69a [56]. Adding to the intrigue is the finding that Obp levels—including those of Obp69a—can be modulated by factors such as exposure to pheromone, feeding or ageing [56–58]. The expression of more than 25 Obps outside the olfactory organs of the fly suggests a multitude of interesting functions.

In summary, Obps are a remarkably numerous and diverse class of proteins found widely among the insect species of the world. Enormous metabolic resources are devoted to their abundant synthesis in the olfactory organs, but their roles in olfaction deserve much greater investigation. Likewise, it seems clear that many other roles in a variety of organs await discovery. Analysis of Obp function *in vivo* is likely to yield a wealth of insight into insect biology.

Data accessibility. This article has no additional data.

Competing interests. We declare we have no competing interests.

Funding. NIH grants DC02714, DC04729, DC11697, and AI115648 to J.R.C. NSF Graduate Research Fellowship, NIH grant T32 GM007499, Dwight N. and Noyes D. Clark Scholarship Fund, and a Scholar Award from International Chapter of the P.E.O. Sisterhood to J.S.S.

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
