## [Reviewer comments · Open Biology]

Review History

RSOB-18-0208.R0 (Original submission)

Review form: Reviewer 1

Recommendation

Major revision is needed (please make suggestions in comments)

Are each of the following suitable for general readers?

- a) **Title**
Yes
- b) **Summary**
Yes
- c) **Introduction**
Yes

Is the length of the paper justified?

Yes

Should the paper be seen by a specialist statistical reviewer?

No

Is it clear how to make all supporting data available?

Not Applicable

Is the supplementary material necessary; and if so is it adequate and clear?

Not Applicable

Do you have any ethical concerns with this paper?

No

Comments to the Author

In the review by Sun, Xiao, and Carlson, the authors tackle Odorant Binding Proteins (OBPs)- a class of molecules highly enriched in olfactory tissues, but whose functions are poorly understood. The goal of the review is to summarize what is currently known about OBPs, highlighting recent discoveries made by the authors in *Drosophila*.

A review about OBPs is timely, given the new discoveries by the Carlson lab. It is useful to revisit OBPs in light of these new discoveries. While I am in favor of the review, I think the authors could dig a little deeper in discussing OBPs. I find reviews most useful when they are a synthesis of knowledge from experts in the field; a way to gain insight into a topic from those who have thought long and hard about it. As it is now, the review is mainly cursory, lightly touching upon different topics associated with OBPs. For example, after reading the review, I was still left slightly confused as to what OBPs actually do, which is essentially how I felt before reading the review. As such, it would be helpful (and an opportunity for the authors) to reflect upon the author's current models/thinking for OBPs. How might a beginning graduate student think about this curious class of molecules? I describe sections that could be improved, which might help the authors enrich the review, and make it more impactful.

1. Question. If OBPs can bind many odorants (chemicals?) in vitro assays, can these studies be used to determine if the chemical binding is relevant? Do the authors have recommendations on how to interpret such results?
2. OBPs appear to have diverse functions, likely far beyond binding odorants. Should they be renamed? Does classifying them as odorant binding proteins hinder investigations into their function? Are they enriched in other tissues?
3. It would be interesting to include a phylogenetic tree for OBPs (for the OBPs in *Drosophila*, and include OBPs from other insects if possible), and then highlight which OBPs have known in vivo functions (ie, LUSH for cVA pheromone binding, OBP28a as quencher, OBP59a as humidity sensor). Would this reveal that there are sub-families? That some OBPs might be pheromone binding proteins (as previously implicated), others more similar to OBP59a may act to interact with non-odorants, etc? Do the authors think that OBPs as a whole might be better classified into smaller protein families? If so, what should be the classifications?
4. The potential function of OBPs in non-model insects could be expanded. For example, ticks do not contain ORCO/Ors but do contain OBPs. It is suspected that OBPs might act directly in odorant detection (see work by Roe and Sonenshine). The author's insights on this would be of value. In addition, OBP1 is suspected to play a large role in odorant and repellent detection in

mosquitoes (see: Biessmann Plos one 2010, Qiao Cell Mol Life Sci 2010, Tsitsanou Cell Mol Life Sciences 2011, among others).

5. Section 2 of the review. Please clarify how OBPs are classified from an insect's genome (ie, Agam has 69, cockroach has 109). If they are diverse in sequence, what hallmarks are used to determine if a protein is an OBP? Could this classification be too broad?
6. Figure 2. Please point out the three disulfide bridges on the crystal structure. As these bridges are a hallmark of an OBP, it will be useful to see them presented in the structure. Similarly, please label the 6 alpha-helices in the crystal structure (e.g., different shades of color, or by numbers). If one can't see all 6 helices in a single view, maybe include an image of it rotated 90°.
7. The crystal structure of Agam OBP1 is hypothesized to be a dimer (Fig 3 in Tsitsanou 2012). How does this relate to the crystal structure shown by the authors in Fig 2? Can OBPs function as both monomers and dimers?
8. Figure 4. Perhaps shade the lymph region to emphasize that it represents an aqueous region.
9. Are there any general principles for the types of chemicals that OBPs tend to bind? If binding is verified, is it usually to fatty lipids like cuticular pheromones? Mostly hydrophobic chemicals? Have in vitro studies similarly generated the types of chemicals that can be bound to OBPs? Or those that would not be bound?
10. What are the 'tuning' properties of OBPs? Do they tend to bind to certain classes of molecules? Do they tend to be narrowly or broadly tuned?
11. It would be of value if the authors commented on what they view as the future of OBP research. Do the authors think OBPs will be involved in even more functions? If so, what? What is the best way forward? Should efforts be spent performing in vitro binding assays? Or instead on in vivo studies? Would exogenous expression screens be of value? Or only knock-out studies?

Review form: Reviewer 2

Recommendation

Accept with minor revision (please list in comments)

Are each of the following suitable for general readers?

- a) **Title**
Yes
- b) **Summary**
Yes
- c) **Introduction**
Yes

Is the length of the paper justified?

Yes

Should the paper be seen by a specialist statistical reviewer?

No

Is it clear how to make all supporting data available?

Not Applicable

Is the supplementary material necessary; and if so is it adequate and clear?

Not Applicable

Do you have any ethical concerns with this paper?

No

Comments to the Author

The authors have provided a concise review of the insect OBPs, showing how several studies, most decisively from their lab, reveal that the now-standard supposition about their function (transport of hydrophobic ligands from the air to the odorant receptors), is unlikely to be their primary role.

The review is well written and in my opinion describes well the state of the field, which these authors and a few others have recently questioned with strong datasets. As they note, there are innumerable studies of OBPs in insects, their diversity, expression, and ligand-binding, but almost no decisive studies of their function based on the gold standard of knocking out the gene encoding the OBP. In part this is because until very recently this was difficult to do in non-Drosophila insects, hence their and other studies on Drosophila OBP mutants are so critical. This review will go a long way to pushing others in this field to take a more critical view of OBP function, and hopefully will recruit some outside the immediate field to study these abundant, diverse, and clearly very interesting but mostly enigmatic proteins.

I only have a few suggestions for improvement.

First, in the title and the first sentence of the abstract, I encourage the authors to include the word "insect", both to avoid confusion with the mammalian OBPs (which, as they note, are completely different proteins), and any other animals with OBP-like proteins, as with a few unclear exceptions, so far OBPs have only been found in insects and their closest relatives, other hexapods.

Second, I think the authors could include reference to the Gomez-Diaz et al paper from 2013 in PLoS Biology on the role of Obp67d in cVA perception in Drosophila showing that the model of OBP configuration changing upon ligand binding and this change affecting the relevant OR is not correct (something mentioned briefly in line109).

Decision letter (RSOB-18-0208.R0)

22-Nov-2018

Dear Dr Carlson

We are pleased to inform you that your manuscript RSOB-18-0208 entitled "The diverse small proteins called Odorant Binding Proteins" has been accepted by the Editor for publication in

Open Biology. The reviewer(s) have recommended publication, but also suggest some minor revisions to your manuscript. Therefore, we invite you to respond to the reviewer(s)' comments and revise your manuscript.

Please submit the revised version of your manuscript within 14 days. If you do not think you will be able to meet this date please let us know immediately and we can extend this deadline for you.

- 1) A text file of the manuscript (doc, txt, rtf or tex), including the references, tables (including captions) and figure captions. Please remove any tracked changes from the text before submission. PDF files are not an accepted format for the "Main Document".
- 2) A separate electronic file of each figure (tiff, EPS or print-quality PDF preferred). The format should be produced directly from original creation package, or original software format. Please note that PowerPoint files are not accepted.
- 3) Electronic supplementary material: this should be contained in a separate file from the main text and meet our ESM criteria (see <http://royalsocietypublishing.org/instructions-authors#question5>). All supplementary materials accompanying an accepted article will be treated as in their final form. They will be published alongside the paper on the journal website and posted on the online figshare repository. Files on figshare will be made available approximately one week before the accompanying article so that the supplementary material can be attributed a unique DOI.

Online supplementary material will also carry the title and description provided during submission, so please ensure these are accurate and informative. Note that the Royal Society will not edit or typeset supplementary material and it will be hosted as provided. Please ensure that the supplementary material includes the paper details (authors, title, journal name, article DOI). Your article DOI will be 10.1098/rsob.2016[last 4 digits of e.g. 10.1098/rsob.20160049].

- 4) A media summary: a short non-technical summary (up to 100 words) of the key findings/importance of your manuscript. Please try to write in simple English, avoid jargon, explain the importance of the topic, outline the main implications and describe why this topic is newsworthy.

Images

Data-Sharing

It is a condition of publication that data supporting your paper are made available. Data should be made available either in the electronic supplementary material or through an appropriate repository. Details of how to access data should be included in your paper. Please see <http://royalsocietypublishing.org/site/authors/policy.xhtml#question6> for more details.

Data accessibility section

Sincerely,

The Open Biology Team
<mailto:openbiology@royalsociety.org>

Editor's Comment:

The referees seem to have some useful comments. Please address those that will be helpful to improve the article.

Reviewer(s)' Comments to Author:

Referee: 1

Comments to the Author(s)

In the review by Sun, Xiao, and Carlson, the authors tackle Odorant Binding Proteins (OBPs)- a class of molecules highly enriched in olfactory tissues, but whose functions are poorly understood. The goal of the review is to summarize what is currently known about OBPs, highlighting recent discoveries made by the authors in *Drosophila*.

A review about OBPs is timely, given the new discoveries by the Carlson lab. It is useful to revisit OBPs in light of these new discoveries. While I am in favor of the review, I think the authors could dig a little deeper in discussing OBPs. I find reviews most useful when they are a synthesis of knowledge from experts in the field; a way to gain insight into a topic from those who have thought long and hard about it. As it is now, the review is mainly cursory, lightly touching upon different topics associated with OBPs. For example, after reading the review, I was still left slightly confused as to what OBPs actually do, which is essentially how I felt before reading the review. As such, it would be helpful (and an opportunity for the authors) to reflect upon the author's current models/thinking for OBPs. How might a beginning graduate student

think about this curious class of molecules? I describe sections that could be improved, which might help the authors enrich the review, and make it more impactful.

1. Question. If OBPs can bind many odorants (chemicals?) in vitro assays, can these studies be used to determine if the chemical binding is relevant? Do the authors have recommendations on how to interpret such results?
2. OBPs appear to have diverse functions, likely far beyond binding odorants. Should they be renamed? Does classifying them as odorant binding proteins hinder investigations into their function? Are they enriched in other tissues?
3. It would be interesting to include a phylogenetic tree for OBPs (for the OBPs in *Drosophila*, and include OBPs from other insects if possible), and then highlight which OBPs have known in vivo functions (ie, LUSH for cVA pheromone binding, OBP28a as quencher, OBP59a as humidity sensor). Would this reveal that there are sub-families? That some OBPs might be pheromone binding proteins (as previously implicated), others more similar to OBP59a may act to interact with non-odorants, etc? Do the authors think that OBPs as a whole might be better classified into smaller protein families? If so, what should be the classifications?
4. The potential function of OBPs in non-model insects could be expanded. For example, ticks do not contain ORCO/Ors but do contain OBPs. It is suspected that OBPs might act directly in odorant detection (see work by Roe and Sonenshine). The author's insights on this would be of value. In addition, OBP1 is suspected to play a large role in odorant and repellent detection in mosquitoes (see: Biessmann Plos one 2010, Qiao Cell Mol Life Sci 2010, Tsitsanou Cell Mol Life Sciences 2011, among others).
5. Section 2 of the review. Please clarify how OBPs are classified from an insect's genome (ie, Agam has 69, cockroach has 109). If they are diverse in sequence, what hallmarks are used to determine if a protein is an OBP? Could this classification be too broad?
6. Figure 2. Please point out the three disulfide bridges on the crystal structure. As these bridges are a hallmark of an OBP, it will be useful to see them presented in the structure. Similarly, please label the 6 alpha-helices in the crystal structure (e.g., different shades of color, or by numbers). If one can't see all 6 helices in a single view, maybe include an image of it rotated 90°.
7. The crystal structure of Agam OBP1 is hypothesized to be a dimer (Fig 3 in Tsitsanou 2012). How does this relate to the crystal structure shown by the authors in Fig 2? Can OBPs function as both monomers and dimers?
8. Figure 4. Perhaps shade the lymph region to emphasize that it represents an aqueous region.
9. Are there any general principles for the types of chemicals that OBPs tend to bind? If binding is verified, is it usually to fatty lipids like cuticular pheromones? Mostly hydrophobic chemicals? Have in vitro studies similarly generated the types of chemicals that can be bound to OBPs? Or those that would not be bound?
10. What are the 'tuning' properties of OBPs? Do they tend to bind to certain classes of molecules? Do they tend to be narrowly or broadly tuned?
11. It would be of value if the authors commented on what they view as the future of OBP research. Do the authors think OBPs will be involved in even more functions? If so, what? What is the best way forward? Should efforts be spent performing in vitro binding assays? Or instead

on in vivo studies? Would exogenous expression screens be of value? Or only knock-out studies?

Referee: 2

Comments to the Author(s)

The authors have provided a concise review of the insect OBPs, showing how several studies, most decisively from their lab, reveal that the now-standard supposition about their function (transport of hydrophobic ligands from the air to the odorant receptors), is unlikely to be their primary role.

The review is well written and in my opinion describes well the state of the field, which these authors and a few others have recently questioned with strong datasets. As they note, there are innumerable studies of OBPs in insects, their diversity, expression, and ligand-binding, but almost no decisive studies of their function based on the gold standard of knocking out the gene encoding the OBP. In part this is because until very recently this was difficult to do in non-Drosophila insects, hence their and other studies on Drosophila OBP mutants are so critical. This review will go a long way to pushing others in this field to take a more critical view of OBP function, and hopefully will recruit some outside the immediate field to study these abundant, diverse, and clearly very interesting but mostly enigmatic proteins.

I only have a few suggestions for improvement.

First, in the title and the first sentence of the abstract, I encourage the authors to include the word "insect", both to avoid confusion with the mammalian OBPs (which, as they note, are completely different proteins), and any other animals with OBP-like proteins, as with a few unclear exceptions, so far OBPs have only been found in insects and their closest relatives, other hexapods.

Second, I think the authors could include reference to the Gomez-Diaz et al paper from 2013 in PLoS Biology on the role of Obp67d in cVA perception in Drosophila showing that the model of OBP configuration changing upon ligand binding and this change affecting the relevant OR is not correct (something mentioned briefly in line109).

Author's Response to Decision Letter for (RSOB-18-0208.R0)

See Appendix A.

Decision letter (RSOB-18-0208.R1)

28-Nov-2018

Dear Dr Carlson

We are pleased to inform you that your manuscript entitled "The diverse small proteins called Odorant Binding Proteins" has been accepted by the Editor for publication in Open Biology.

Sincerely,

The Open Biology Team
mailto: openbiology@royalsociety.org

Appendix A

Yale University

Department of Molecular, Cellular
and Developmental Biology
Yale University
P. O. Box 208103
New Haven, Connecticut 06520-8103

Campus address:
Kline Biology Tower
219 Prospect Street
Telephone: 203 432-3460
Fax: 203 432-6161

11-27-18

Editor
Open Biology

Dear Editor,

We are very grateful to the reviewers for their helpful comments, which we have addressed as follows:

Reviewer 1

1. *Question. If OBPs can bind many odorants (chemicals?) in vitro assays, can these studies be used to determine if the chemical binding is relevant? Do the authors have recommendations on how to interpret such results?*

--We have added to the ms. a statement that "binding affinities *in vitro* may not always reflect binding affinities *in vivo*, where conditions may differ. *In vitro* studies are extremely useful in making predictions about Obp function *in vivo*, but these predictions need to be tested."

2. *OBPs appear to have diverse functions, likely far beyond binding odorants. Should they be renamed? Does classifying them as odorant binding proteins hinder investigations into their function? Are they enriched in other tissues?*

--We think it would be better to rename Obps after we know more about their functions. We have added a statement that "In fact, recent RNAseq data in *Drosophila* has revealed expression of more than 25 *Obps* in tissues outside the olfactory organs of the fly."

3. *It would be interesting to include a phylogenetic tree for OBPs (for the OBPs in *Drosophila*, and include OBPs from other insects if possible), and then highlight which OBPs have known *in vivo* functions (ie, LUSH for cVA pheromone binding, OBP28a as quencher, OBP59a as humidity sensor).*

Would this reveal that there are sub-families? That some OBPs might be pheromone binding proteins (as previously implicated), others more similar to OBP59a may act to interact with non-odorants, etc? Do the authors think

that OBPs as a whole might be better classified into smaller protein families? If so, what should be the classifications?

--We have examined a published phylogenetic tree of Obps and have not found a compelling relationship between position on the tree and function, nor a compelling reason to reclassify them. We have, however, added a statement that the tree includes not only classical Obps but also ones with more or fewer cysteine residues.

4. The potential function of OBPs in non-model insects could be expanded. For example, ticks do not contain ORCO/Ors but do contain OBPs. It is suspected that OBPs might act directly in odorant detection (see work by Roe and Soneshine). The author's insights on this would be of value. In addition, OBPI is suspected to play a large role in odorant and repellent detection in mosquitoes (see: Biessmann Plos one 2010, Qiao Cell Mol Life Sci 2010, Tsitsanou Cell Mol Life Sciences 2011, among others).

--We consider the identification of Obp-like proteins in ticks to be tentative and prefer not to emphasize them. However, the revised ms. contains a reference to Biessman *PLoS One* 2010.

5. Section 2 of the review. Please clarify how OBPs are classified from an insect's genome (ie, Agam has 69, cockroach has 109). If they are diverse in sequence, what hallmarks are used to determine if a protein is an OBP? Could this classification be too broad?

--We have explained how the 109 Obps were identified and acknowledged that these numbers include some variants of the canonical structure. We have also indicated that the abundantly expressed antennal Obps are all of the classical structure.

6. Figure 2. Please point out the three disulfide bridges on the crystal structure. As these bridges are a hallmark of an OBP, it will be useful to see them presented in the structure.

Similarly, please label the 6 alpha-helices in the crystal structure (e.g., different shades of color, or by numbers). If one can't see all 6 helices in a single view, maybe include an image of it rotated 90°.

--We have replaced the original figure with a figure that shows the disulfide bridges and that labels the 6 alpha-helices.

7. The crystal structure of Agam OBPI is hypothesized to be a dimer (Fig 3 in Tsitsanou 2012). How does this relate to the crystal structure shown by the authors in Fig 2? Can OBPs function as both monomers and dimers?

--We have added a statement indicating that "Crystallized Obps have revealed a dimeric structure with a binding pocket that consists of a tunnel extending into each of the two subunits (Leite 2009, Wogulis 2006)".

8. *Figure 4. Perhaps shade the lymph region to emphasize that it represents an aqueous region.*

--Amended as suggested; we thank the reviewer for this excellent suggestion.

9. *Are there any general principles for the types of chemicals that OBPs tend to bind? If binding is verified, is it usually to fatty lipids like cuticular pheromones? Mostly hydrophobic chemicals? Have in vitro studies similarly generated the types of chemicals that can be bound to OBPs? Or those that would not be bound?*

--We have added a statement indicating that "Most Obps bind hydrophobic compounds (Pelosi CMLS 06)"

10. *What are the 'tuning' properties of OBPs? Do they tend to bind to certain classes of molecules? Do they tend to be narrowly or broadly tuned?*

--We have added a statement that "some appear to bind a broad spectrum of hydrophobic ligands (Li PLOS One 16)."

11. *It would be of value if the authors commented on what they view as the future of OBP research. Do the authors think OBPs will be involved in even more functions? If so, what? What is the best way forward? Should efforts be spent performing in vitro binding assays? Or instead on in vivo studies? Would exogenous expression screens be of value? Or only knock-out studies?*

--We have rewritten the last section of the paper to provide a more explicit vision of a future for the field. The revised section reflects our view that more functions will be found, and that *in vivo* analysis is the most promising path forward. We suggest both physiological and anatomical avenues for research.

Referee: 2

1. *First, in the title and the first sentence of the abstract, I encourage the authors to include the word "insect", both to avoid confusion with the mammalian OBPs (which, as they note, are completely different proteins), and any other animals with OBP-like proteins, as with a few unclear exceptions, so far OBPs have only been found in insects and their closest relatives, other hexapods.*

--We have added the word "insect" to the first sentence of the Abstract.

2. *Second, I think the authors could include reference to the Gomez-Diaz et al paper from 2013 in PLoS Biology on the role of Obp67d in cVA perception in*

Drosophila showing that the model of OBP configuration changing upon ligand binding and this change affecting the relevant OR is not correct (something mentioned briefly in line 109).

--The original version of our ms. was not clear in line 109. Gomez-Diaz refuted a model from Dean Smith's lab in which odor receptor Or67d is activated directly by an Obp bound to cVA. In line 109 we were referring to a model of Walter Leal in which other Obps change configuration due to interaction with a charged membrane. We have reworded this line to make our intent clearer.

Thanks very much for your consideration,

John Carlson
Higgins Professor